# Comparing Vaginal and Endometrial Microbiota Using Culturomics: Proof of Concept

**DOI:** 10.3390/ijms24065947

**Published:** 2023-03-21

**Authors:** Robin Vanstokstraeten, Ellen Callewaert, Susanne Blotwijk, Eleni Rombauts, Florence Crombé, Kristof Emmerechts, Oriane Soetens, Kristof Vandoorslaer, Deborah De Geyter, Camille Allonsius, Leonore Vander Donck, Christophe Blockeel, Ingrid Wybo, Denis Piérard, Thomas Demuyser, Shari Mackens

**Affiliations:** 1Department of Microbiology and Infection Control, Vrije Universiteit Brussel (VUB), Universitair Ziekenhuis Brussel (UZ Brussel), 1090 Brussels, Belgium; 2Department of Pharmaceutical Sciences, Entity of In Vitro Toxicology, Vrije Universiteit Brussel (VUB), Laarbeeklaan 103, 1090 Brussels, Belgium; 3Biostatistics and Medical Informatics Research Group (BISI), Vrije Universiteit Brussel (VUB), Laarbeeklaan 103, 1090 Brussels, Belgium; 4Department of Bioscience Engineering, University of Antwerp (UA), 2020 Antwerp, Belgium; 5Brussels IVF, Universitair Ziekenhuis Brussel (UZ Brussel), 1090 Brussels, Belgium; 6AIMS Lab, Center for Neurosciences, Faculty of Medicine and Pharmacy, Vrije Universiteit Brussel (VUB), Laarbeeklaan 103, 1090 Brussels, Belgium

**Keywords:** culturomics, endometrial microbiome, vaginal microbiome, MALDI-TOF, 16S rRNA, ART, embryo implantation

## Abstract

It is generally accepted that microorganisms can colonize a non-pathological endometrium. However, in a clinical setting, endometrial samples are always collected by passing through the vaginal–cervical route. As such, the vaginal and cervical microbiomes can easily cross-contaminate endometrial samples, resulting in a biased representation of the endometrial microbiome. This makes it difficult to demonstrate that the endometrial microbiome is not merely a reflection of contamination originating from sampling. Therefore, we investigated to what extent the endometrial microbiome corresponds to that of the vagina, applying culturomics on paired vaginal and endometrial samples. Culturomics could give novel insights into the microbiome of the female genital tract, as it overcomes sequencing-related bias. Ten subfertile women undergoing diagnostic hysteroscopy and endometrial biopsy were included. An additional vaginal swab was taken from each participant right before hysteroscopy. Both endometrial biopsies and vaginal swabs were analyzed using our previously described WASPLab-assisted culturomics protocol. In total, 101 bacterial and two fungal species were identified among these 10 patients. Fifty-six species were found in endometrial biopsies and 90 were found in vaginal swabs. On average, 28 % of species were found in both the endometrial biopsy and vaginal swab of a given patient. Of the 56 species found in the endometrial biopsies, 13 were not found in the vaginal swabs. Of the 90 species found in vaginal swabs, 47 were not found in the endometrium. Our culturomics-based approach sheds a different light on the current understanding of the endometrial microbiome. The data suggest the potential existence of a unique endometrial microbiome that is not merely a presentation of cross-contamination derived from sampling. However, we cannot exclude cross-contamination completely. In addition, we observe that the microbiome of the vagina is richer in species than that of the endometrium, which contradicts the current sequence-based literature.

## 1. Introduction

The female genital tract consists of the vagina, cervix, uterus, fallopian tubes, and ovaries. Based on both targeted and shotgun metagenomics data, it appears that different microbial communities colonize each site. The lower genital tract is typically colonized with a high microbial load of *Lactobacillus* species, whereas the upper genital tract is colonized with a low microbial load (1000–10,000 times lower) and a wide variety of species. *Firmicutes*, *Proteobacteria*, *Actinobacteria*, and *Bacteroidetes* are the main phyla described at the uterine level by sequence-based studies [1].

Several studies suggest a relationship between a healthy, eubiotic vaginal microbiome and the occurrence of certain *Lactobacillus* species [2,3,4]. The predominance of *L. crispatus*, *L. gasseri*, or *L. jensenii* is associated with a eubiotic state, whereas the predominance of *L. iners* tends to be associated with a transitional state, characterized by more diverse microbiota. Most *Lactobacillus* species can produce D- and L-lactic acid, whereas *L. iners* can only produce L-lactic acid. The inhibitory effects of D-lactic acid are considerably more potent than the inhibitory effects of L-lactic acid, which explains the susceptibility to microbial changes within *L. iners*-dominated microbiota. Furthermore, *L. iners* is active in a wider pH range (pH > 4.5) compared to the other *Lactobacillus* species (pH ≈ 4.0) [5]. Although *L. iners*-dominated microbiota are associated with more diverse microbiota, it is very prevalent in complaint-free women, suggesting that this microorganism could be a friend and not merely a foe [6]. The overgrowth of anaerobic bacteria, such as *Gardnerella* species, represents a dysbiotic vaginal microbiome, which is often associated with pathological conditions, such as bacterial vaginosis, and adverse reproductive outcomes [7,8]. Although *Gardnerella* is thus often considered a pathobiont in the vaginal niche, it is the dominant genus in a major part of the healthy Western-European population [6]. It has been suggested that the association between *Gardnerella* and disease might depend on the specific strains and species [9,10].

Next to the vaginal microbiota, the endometrial microbiome is also potentially associated with certain pathologies and human reproductive efficiency (e.g., endometriosis, chronic endometritis, embryonic implantation, conception, (sub)fertility, and pregnancy outcomes [11,12,13]. A study by *Moreno* et al. suggests that a *Lactobacillus*-dominated endometrium is linked to better reproductive outcomes [14]. In contrast, *Hashimoto* et al. concluded that reproductive outcomes are comparable between non-*Lactobacillus*-dominated and *Lactobacillus*-dominated endometrial microbiomes [15]. It has to be acknowledged that analysis of the endometrial microbiome is hampered by multiple technical limitations and the low accessibility to healthy controls. Sequencing bias, originating from DNA/RNA contamination in laboratory reagents and extraction kits, is a major drawback in sequence-based studies, especially when studying low-biomass microbiota like that of the endometrium [16,17,18]. Sequencing of microbiota often gives only genus-level identifications with no information on which species are present. Extracting and sequencing DNA without RNA also does not allow distinguishing between living microorganisms and genetic fragments. Therefore, applying and validating alternative methodologies to overcome these biases is crucial for further understanding. Another major limitation when investigating the endometrial microbiome is the possible cross-contamination with microorganisms originating from the lower reproductive tract [18,19]. To take a biopsy from the endometrium in a clinical setting always implies crossing the vagina and cervix, even when working with a sterile inner–outer catheter. It is therefore of interest to study paired samples from both anatomical sites with a similar methodology to explore the existence of a unique endometrial microbiome that differs from that of the vagina. Moreno et al. studied this issue in 2016 using paired vaginal and endometrial samples and suggested, based on sequencing data, a differential endometrial microbiome [14]. However, since the biomass in the uterine cavity and endometrium is several thousand times smaller than that of the vagina, these analyses are at high risk of DNA/RNA contamination from laboratory reagents and extraction kits [16]. In addition, the microbiome of the vagina is probably much more prone to depth bias, as the signal of highly abundant taxa, such as *Lactobacilli*, might swamp the signal of less abundant taxa [20].

In order to overcome the above-mentioned, sequence-related shortcomings, we investigated culturomics as a new and alternative culture-based methodology to explore the endometrial microbiome at the species level [21]. Culturomics is a high-throughput culture methodology, combining different agar plates, enrichment broths, and (an)aerobic incubation conditions to cultivate virtually all viable microbiota. Pure colonies of these cultured species are then identified by matrix-assisted laser desorption/ionization–time-of-flight mass spectrometry (MALDI-TOF MS) or full-gene 16S rRNA sequencing [22]. Applying this methodology, we previously reported a unique insight into the endometrial microbiome as we identified 85 different microorganisms in 10 endometrial biopsies, of which 53 were described for the first time in the endometrium [21].

In the current study, we used culturomics to investigate to what extent the endometrial microbiome matches that of the vagina. Paired sampling of vaginal swabs and endometrial biopsies was done for ten patients. As such, we are the first to make this paired comparison based on culturomics without the results being affected by sequencing bias.

## 2. Results

Ten endometrial biopsies and ten vaginal swabs were included in this study. The clinical characteristics of the women are depicted in Table 1. A total of 4782 colonies were identified using the MALDI Biotyper system. Overall, we identified 101 bacterial and two fungal (*Candida albicans* and *Nannizzia incurvata*) species. Twenty-seven of these bacterial isolates (26.73%) were Gram-negatives and 41 (40.59%) were obligate anaerobes. These 103 species belonged to 54 different genera and 36 different families. Seven species (*Actinomyces odontolyticus*, *Jonquetella anthropi*, *Ligilactobacillus salivarius*, *Limosilactobacillus fermentum*, *Lactobacillus mulieris*, *Murdochiella vaginalis* and *Peptoniphilus harei*) were not identifiable with MALDI-TOF MS and were identified using 16S rRNA gene sequencing. All the identified species and genera are summarized in Figure 1. At the species level, we observed unique microbiota among all the patients. This reflects in the fact that 45 of the 103 species (43.69%) were discovered in only one of the 10 paired samples (Figure 1). These results suggest that the number of species would probably increase strongly if the study population were expanded.

### 2.1. Comparing the Endometrial and Vaginal Microbiota

In total, fifty-six species were found in the endometrial biopsies and 90 were found in the vaginal swabs. Forty-six were identified uniquely in the vagina and 13 in the endometrium. The genera *Campylobacter, Mobiluncus, Nannizzia, Ralstonia, Stenotrophomonas,* and *Varibaculum* were found only in the endometrium (Figure 1 and Figure 2). The concordance of species between the vaginal and endometrial microbiota ((species found in both vagina and endometrium/all species found) * 100) differed from 6.25 to 47.06% (Figure 3). Since culturomics does not provide abundance data, alpha diversity can only be measured in terms of richness, in our case the number of species. We observed significantly more (*p* = 0.002) species in the vagina than in the endometrium (Figure 4). The microbiota of paired samples seem to be associated with each other, as analysis of UniFrac distances between all 20 samples with a permutation test resulted in significantly smaller (*p* = 0.005) UniFrac distances between paired endometrial–vaginal samples than all other combinations.

### 2.2. Lactobacillus Species

All endometrial and vaginal samples harbored one or more *Lactobacillus*, *Ligilactobacillus*, or *Limosilactobacillus* species. Ten different species were identified within these genera. Despite the sample size being too small to draw any statistically supported conclusions about this, the presence of specific species within these genera seems to potentially influence the alpha diversity of these microbiota, particularly that of the vagina. Scatterplots of *L. jensenii* and *L. iners* highlight these interesting findings, as *L. jensenii* appeared in the low-diversity microbiome profiles and *L. iners* in the high-diversity microbiome profiles (Appendix A). In the most diverse vaginal microbiome profile, *L. iners* was the only *Lactobacillus* species present. This microbiome harbored 45 species, of which multiple dysbiosis-associated species like *Gardnerella vaginalis*, *Candida albicans*, *Fusobacterium nucleatum,* and multiple anaerobes belonging to the genera *Prevotella* and *Veillonella.*

## 3. Discussion

We applied culturomics to 10 paired vaginal swabs and endometrial biopsies. We observed important differences in the paired microbiota of all 10 patients with a mean concordance between the vaginal and endometrium microbiota of 28.6%. We added 20 species to the culturomics-generated endometrial species list from our previous study, now describing 105 different microorganisms detected in the endometrium with this specific technique [21]. Based on the species already described in the literature, we detected 17 of these 20 species for the first time in the endometrial microbiome, highlighting the more in-depth data generated by culture-based microbiota studies [2,21,23]. Although this is not the first comparative study between endometrial and vaginal microbiota, it is the first time that a culturomics approach has been performed. Moreno et al. previously compared the endometrial and vaginal microbiota in 13 fertile women using sequencing. They observed different bacterial communities between both anatomical sites, suggesting the existence of a unique endometrial microbiome [14]. As in the current study, they observed high inter-patient variability and *Lactobacillus* as the most prevalent genus. Although sequencing is a powerful and widely accepted tool, we must approach the results and conclusions of Moreno et al. with great caution, as the biomass of the endometrial microbiome is much lower than that of the vagina. As a result, the sequence-based microbiome of the endometrium is much more sensitive to DNA/RNA contamination. Contaminating DNA and RNA, both human and environmental, could therefore be dominant in low-biomass environments; this can distort the taxonomic distributions and frequencies observed in the endometrial dataset and hamper a valid comparison with the vaginal dataset [16,17]. Although host-depletion and data correction guided by negative controls could partially account for this issue, it is a common misconception this is sufficient to correct for all sequence-related contaminants [18]. An extended culture-based approach, such as the one we used, could overcome these sequencing biases completely, providing an additional tool to reflect potential microbiome differences between both female reproductive tract sites. However, the culturomics approach is also not without shortcomings as it is very restrictive in terms of time, workload, cost, and standardization. Furthermore, we have to take into account ‘non-cultivable’ species and morphological virtually indistinguishable colonies, such as some *Prevotella* and *Peptinophilus* species [20,22,24]. Finally, our approach does not consider the presence of viruses. However, the relevance and function of most viruses in human microbiota, and more specifically the female genital tract, are poorly understood [25].

To our knowledge, all the previous sequence-based studies describe a greater diversity of species in the endometrium than in the vagina [1]. It has been suggested that the lower pH in the vagina, compared to the endometrium, harbors a less-favorable niche for most microorganisms [26]. In contrast to previous literature, we identified, in all patients, a higher number of species in the vagina compared to the endometrium. This observation was found to be statistically significant (*p* = 0.002). These discordant results could be explained by the higher influence of DNA/RNA contamination in the endometrial sequence-based datasets. The introduction of contaminating microbial DNA during sample preparation originating from molecular biology grade water, PCR reagents, and DNA extraction kits may swamp the low amount of starting material and generate misleading results. This type of contamination is a concern for both targeted approaches using PCR, and shotgun approaches without the use of PCR [16,17]. A richer vaginal microbiome also seems logical, as the endometrium is less directly exposed to microorganisms compared to the vagina. Despite vagino-uterine contractions leading sperm (and possibly microorganisms too) to the upper genital tract, medicinal manipulations, and hematological spread might cause an influx of microorganisms seeding the endometrium; the endometrium is, anatomically, fairly isolated compared to the vagina, which is in direct contact with the environment [27,28,29].

Several sequence-based studies suggest that the presence and dominance of certain Lactobacilli may play a role in the diversity of the vaginal microbiome. According to most of the currently available literature, we can distinguish several ‘community state types’ of the vaginal microbiome. Although recent sequence-based studies suggest that there is no correlation between the dominance of *L. iners* and a dysbiotic vaginal microbiome, a community state type dominated by *L. iners* could be associated with a higher vaginal pH, characterized by more diverse microbiota [6]. In contrast, community state types dominated by *L. crispatus*, *L. gasseri*, and *L. jensenii* could be associated with a lower vaginal pH, characterized by a less diverse microbiome [30]. Despite our small sample size, the fact that culturomics only provides qualitative data, and that this topic was not the primary interest of this proof of concept, we observe a similar trend: the presence of specific *Lactobacillus* species within these genera seems likely to influence the alpha diversity of these microbiota, particularly that of the vagina. Additional studies based on both culture and sequencing should clarify this.

In conclusion, there is a high similarity between the vaginal and endometrial microbiota based on culturomics performed on paired samples. It is not clear whether the detected differences are real or related to the methodology and potential cross-contamination. Nevertheless, as we found unique species in both the vaginal and endometrial microbiota with concordances ranging between 6.25% and 47.06% among the paired samples, a unique endometrial microbiome could exist and be of clinical relevance.

## 4. Materials and Methods

### 4.1. Setting and Study Design

Ten subfertile women, undergoing diagnostic hysteroscopy followed by endometrial biopsy as part of a routine work-up at Brussels IVF, Universitair Ziekenhuis Brussel, were included in this study after their informed consent. Right before the hysteroscopy, a vaginal swab was taken. After hysteroscopy, an endometrial biopsy was obtained with a Pipelle de Cornier. No disinfection was performed. The endometrial biopsy and the vaginal swab were collected and transported within minutes to the microbiology laboratory in an eSwab tube (Copan Diagnostics, Brescia, Italy) to minimalize the possible death of obligate anaerobe microorganisms [31].

### 4.2. Culturomics

Culturomics was performed as described in our previous proof-of-concept study for both the endometrial biopsy and the vaginal swab [21]. In that reference, a very detailed description and schematic overviews of the used culture protocols, the composition of the culture media and additives, the targeted species, sterility protocols, and WASP settings are available. Briefly, endometrial biopsies and vaginal swabs were used to culture microbiota for up to 30 days in multiple enriched aerobic and anaerobic conditions. Subsequent WASPLab-assisted culturomics enabled a standardized methodology, with high traceability and reproducibility. Matrix-assisted laser desorption/ionization–time-of-flight mass spectrometry (MALDI-TOF MS) or full-gene 16S rRNA sequencing was applied to identify all bacterial and fungal isolates.

#### 4.2.1. Direct Inoculation

Before homogenizing the vaginal and endometrial samples, the sample volume was increased by adding 1.0 mL of sterile 0.9% NaCl to each sample in a laminar flow cabinet. Vaginal samples were homogenized using a vortex and endometrial samples were homogenized using a sterile pestle and mortar. After homogenization, the first part of the sample was used for direct inoculation. The following culture media were used: aerobic blood agar, anaerobic agar, chocolate agar PolyViteX VCAT3, MacConkey agar, selective anaerobic agar, Sabouraud agar, Schaedler agar, *Ureaplasma/Mycoplasma* agar, *Ureaplasma* broth, and *Mycoplasma* broth. Using the five-fold streaking pattern, 30 µL of the sample was manually inoculated per agar plate. Two drops of the sample were used per broth. The culture media were incubated for five days in conventional aerobic and anaerobic incubators at 37 °C. All morphologically different colonies were identified using MALDI-TOF MS. If the MALDI-TOF MS did not succeed in identifying the strain, full 16S rRNA gene sequencing was performed on a pure colony of that strain. The broths were assessed visually: a green color in the *Ureaplasma* broth indicated the growth of the *Ureaplasma* species, and a red color in the *Mycoplasma* broth indicated the growth of *Mycoplasma*.

#### 4.2.2. Pre-Incubation

An aerobic and anaerobic blood culture bottle (BACT ALERT FN and FA Plus, Marcy-l’Etoile, France) enriched with 2.0 mL sterile rumen fluid, 2.0 mL sterile sheep blood, and 2.0 mL sterile homemade supplement mix were used to pre-incubate the second part of each homogenized sample. Using sterile needles and syringes, both enriched blood culture bottles were inoculated with 0.5 mL of sample working in a laminar flow cabinet. These enriched and inoculated blood culture bottles were incubated at 37 °C. After 1, 5, 10, and 30 days post-incubation, 1 mL of pre-incubated medium was transferred to a Vacuette tube (Greiner, Alphen aan de Rijn, The Netherlands). Together with the appropriate agar plates, this tube was sent to WASPLab (Copan Diagnostics, Brescia, Italy) for automatic inoculation using a 1.0 µL loop and the five-fold streaking pattern. The following agar plates were inoculated automatically using WASPLab: aerobic blood agar, anaerobic agar, chocolate agar PolyViteX VCAT3, MacConkey agar, selective anaerobic agar, Sabouraud agar, and Schaedler agar. Aerobic agars were automatically incubated in CO_2_ and non-CO_2_ aerobic incubators provided by WASPLab. However, as WASPLab does not feature an anaerobic incubator yet, anaerobic agars were placed manually in our conventional aerobic incubator immediately after inoculating with WASPLab. The culture media were incubated for five days at 37 °C. All morphologically different colonies were identified using MALDI-TOF MS. If the MALDI-TOF MS did not succeed in identifying the strain at the species level, full 16S rRNA gene sequencing was performed on a pure colony of that strain. To ensure sterility of the used substances, a blank aerobic and anaerobic blood culture bottle enriched with 2.0 mL sterile rumen fluid, 2.0 mL sterile sheep blood, and 2.0 mL sterile homemade supplement mix accompanied every sample throughout the complete process.

#### 4.2.3. Identification of the Colonies

A MALDI-TOF MS Biotyper Sirius (Bruker Daltonics, Bremen, Germany) system equipped with the Smartbeam MBT version GFLIC-2 laser-positive mode was used to identify all morphologically different colonies. These pure colonies were spotted on a MALDI MSP 96 polished steel target (Bruker Daltonics, Bremen, Germany) after which they were overlaid with 1 µL formic acid and dried at room temperature. Then, the spots were overlaid with 1 µL matrix solution of α-cyano-4-hydroxycinnamic acid (Bruker Daltonics, Bremen, Germany), 10 mg/mL in standard solvent solution (50% acetonitrile, 47.5% water, and 2.5% trifluoroacetic acid), and once again dried at room temperature. The MBT-AutoX method was used to generate the spectra: 240 shots in one spectrum, attenuator offset = 54%, attenuator range = 20%, initial laser power 30%, maximum laser power 40%, and frequency = 200 Hz. The manufacturer’s software FlexControl version 3.4 build 207.20 was used to collect the spectra. The collected spectra were analyzed using the MALDI MBT compass 4.1 build 100. Isolates with a unique hit and an identification score above 2.0 were considered accurately identified and isolates with an identification score below 2.0 were considered not identifiable using MALDI-TOF MS. Eurofins Genomics performed full gene 16S rRNA sequencing on seven isolates with an identification score below 2.0. BioNumerics v.8.1 (Applied Maths, Sint-Matens-Latem, Belgium) was used to assemble the sequenced data. Finally, the Basic Local Alignment Search Tool (BLAST) available at the U.S. National Library of Medicine (https://blast.ncbi.nlm.nih.gov/ accessed on 20 December 2022) was used for analyzing the assembled data. At least 99.0% identity with a known sequence and ≥0.8% difference with the second-best match was needed to identify these isolates at the species level [32].

### 4.3. Statistics

The microbial alpha diversity of the endometrium was compared with that of the vagina using a Wilcoxon signed rank test. Unweighted UniFrac distances were analyzed using a permutation test in order to show a within-patient correlation between endometrial and vaginal microbiome in terms of beta diversity.

## 5. Conclusions

Our culturomics-based approach generates further insights into the endometrial microbiome. Based on our culturomics-analysed paired samples of the vagina and endometrium, the current data suggest that a unique endometrial microbiome could exist which is not merely a representation of cross-contamination derived from sampling. In addition, we show that, following culturomics, the vaginal microbiome seems richer in species compared to the endometrium, which is in contrast with currently available sequence-based studies. Culturomics-acquired microbiome profiles and their relationship with endometrial pathologies and in-vitro fertilization success rates could be of interest to in future investigations.

## Figures and Tables

**Figure 1 ijms-24-05947-f001:**
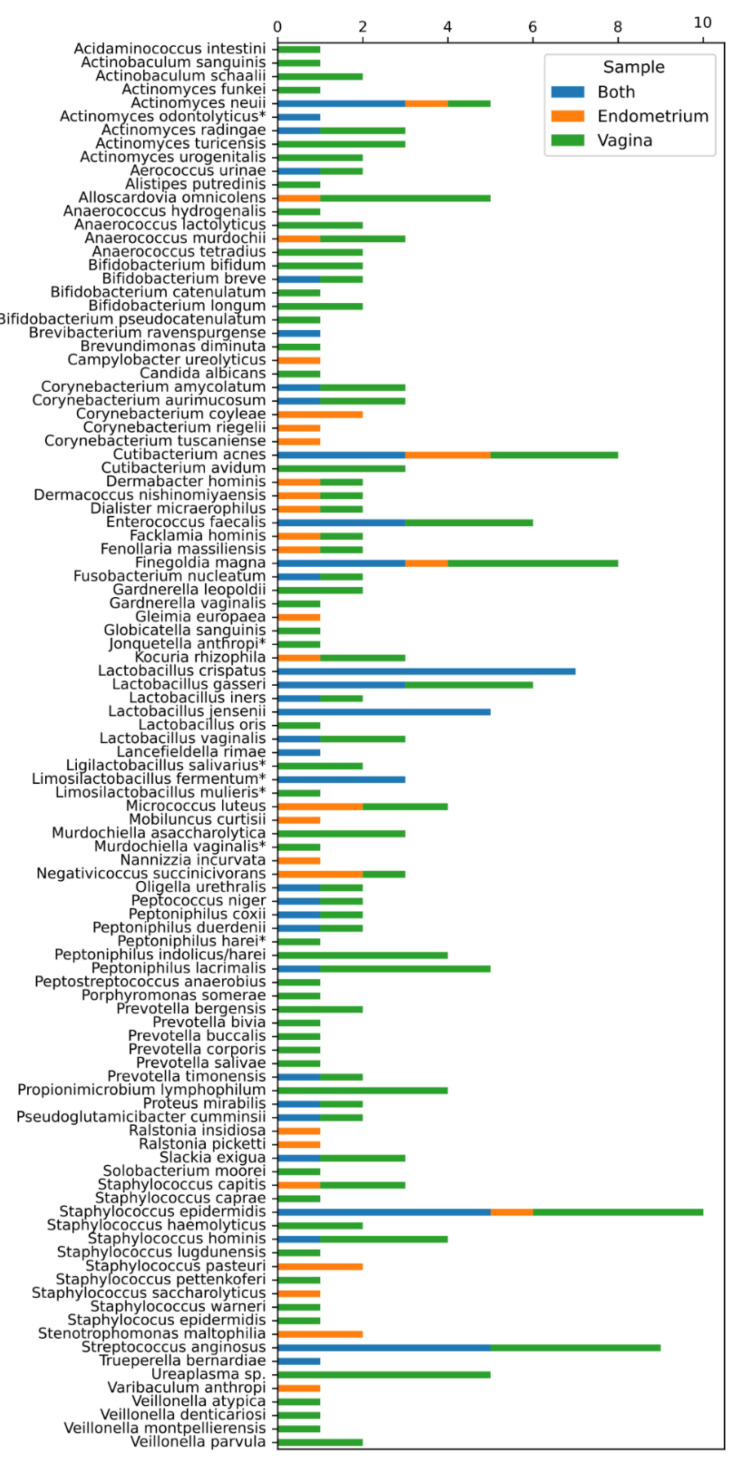
Different microbiota species identified in the 10 patients enrolled in the study. On the y-axis, a summary of the 103 different species found in the 10 endometrial biopsies and 10 vaginal swabs. On the x-axis, the number of times identified per participant. Orange: identified only in the endometrium. Green: identified only in the vagina. Blue: identified in both endometrium and vagina. ** Species identified using 16S rRNA sequencing*.

**Figure 2 ijms-24-05947-f002:**
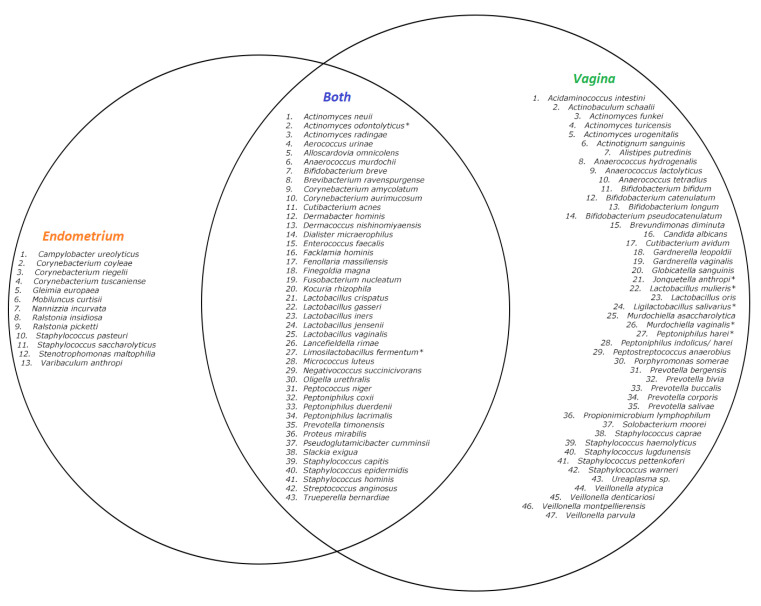
Venn diagram of described species in the 10 samples with culturomics. Left circle: species only found in the endometrium. Right circle: species only found in the vagina. Inner circle: species found in both the vagina and endometrium. ** Species identified using 16S rRNA sequencing*.

**Figure 3 ijms-24-05947-f003:**
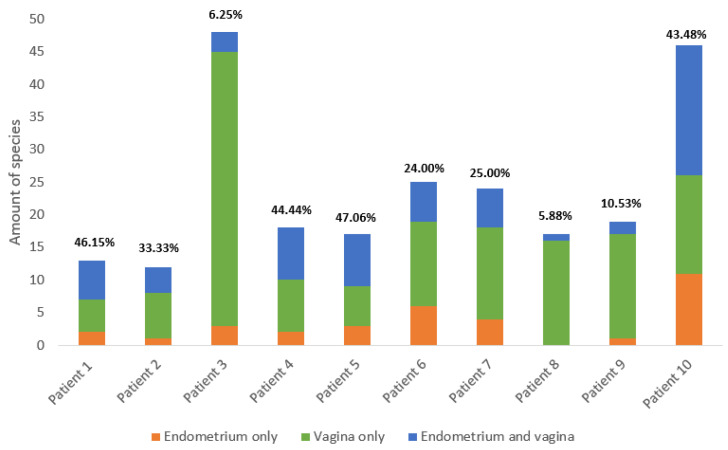
Summary of the number of species in the vagina and endometrium across all 10 patients. The proportion of species identified only in the endometrium is colored orange, the proportion identified only in the vagina is colored green, and the proportion identified in both vagina and endometrium is colored blue. The concordance ((species found in both vagina and endometrium/all species found) * 100) of species between the vaginal and endometrial microbiota is given above the corresponding bars. Detailed information per patient is available in the Appendix A.

**Figure 4 ijms-24-05947-f004:**
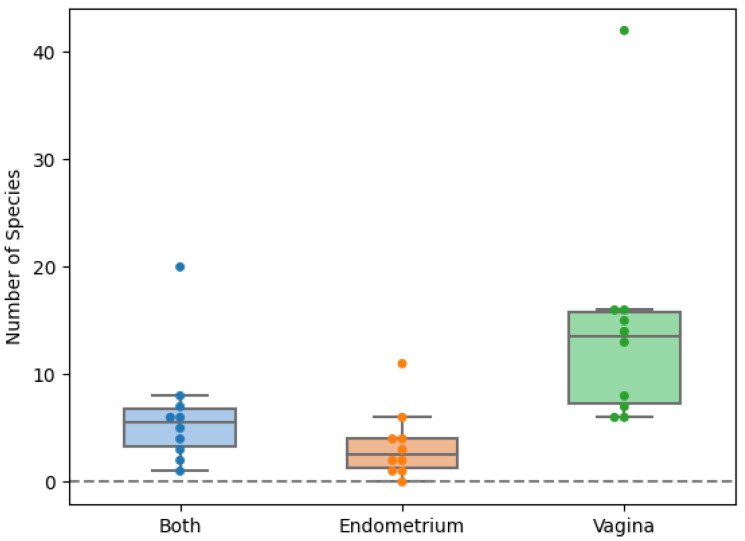
Scattered boxplot for the number of species exclusively found in the vagina, endometrium or in both locations. Orange: exclusively found in the endometrium. Green: exclusively found in the vagina. Blue: found in both vagina and endometrium.

**Table 1 ijms-24-05947-t001:** Clinical data from the 10 patients enrolled in the study. Observations of the vagina, cervix, cavum, and ostium were documented in the context of their fertility trajectory. From every patient, a biopsy of the endometrium was analyzed in the laboratory of anatomopathology for the presence of plasma cells indicating inflammation.

Patient	Age (Years)	Ethnicity	Vagina	Cervix	Cavum	Left Ostium	Right Ostium	Anatomopathology
1	44	Caucasian	Normal	Normal	Normal	Normal	Normal	Normal histology
2	30	Caucasian	Normal	Normal	Normal	Normal	Normal	Normal histology
3	34	Caucasian	Normal	Normal	Normal	Normal	Normal	Normal histology
4	44	Caucasian	Normal	Normal	Normal	Normal	Normal	Isolated plasmacells
5	31	Caucasian	Normal	Normal	Inflammatory	Normal	Normal	Isolated plasmacells
6	35	Caucasian	Normal	Normal	Normal	Normal	Normal	Normal histology
7	41	Caucasian	Normal	Normal	Atrophic	Normal	Normal	Normal histology
8	34	Caucasian	Normal	Normal	Normal	Normal	Normal	Isolated plasmacells
9	36	Caucasian	Normal	Normal	Normal	Normal	Normal	Isolated plasmacells
10	43	Caucasian	Normal	Normal	Normal	Normal	Normal	Isolated plasmacells

## Data Availability

The datasets generated and/or analyzed during the current study are available in the National Library of Medicine repository. GenBank submission: SUB12387136, Prokaryotic 16S rRNA/Prokaryotic 16S rRNA urogenital microbiota, OP968027-OP968039.

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
