# Peer review of "Comparing Vaginal and Endometrial Microbiota Using Culturomics: Proof of Concept"

_ijms, 2023, doi:10.3390/ijms24065947_

Round 1
Reviewer 1 Report
In the article entitled “Comparing the Vaginal and Endometrial Microbiota using Culturomics: proof-of-concept”, the authors compare the endometrial and vaginal microbiota using culturomic methods on paired samples of endometrial biopsies and vaginal smears from 10 subfertile women. The results demonstrate that the endometrial microbiome is unique and distinct from the vaginal microbiome. The study further demonstrates that the endometrial microbiome is less diverse than the vaginal microbiome. The article is well structured and contributes to the knowledge of the microbiota of the female reproductive tract.
The article is a proof of concept whose results contribute to the knowledge of the diversity of the endometrial and uterine microbiome. Although this study was conducted in subfertile patients, the results show differences and similarities in the species of microorganisms found in both anatomical locations. The identification of microorganism species associated with reproductive failure or obstetric complications could be of great clinical relevance. Therefore, the article is relevant and provides the background for future research.
These last results contradict those already published, but the differences between the investigations may be due to the different methodologies used to isolate and identify the microorganisms. In addition, the authors identify new species of microorganisms in the endometrium that have not been described before. The discussion is interesting and the results are appropriately contrasted with the bibliography. Thus, the article contributes to the understanding of the diversity of the microbiome of the female reproductive tract using culturomics-based techniques.
The results are adequately expressed in figures, tables, and diagrams. Supplementary tables with the identification of lactobacillus and L. inners or L. jensenii species in endometrial and vaginal samples are provided.
However, it is important to clarify whether the protocol and the informed consent of the patients were approved by an ethics committee.
Author Response
We thank this reviewer for taking the time and effort to review our manuscript.
The ethical approval is clarified under ‘Institutional Review Board Statement’: The study was conducted in accordance with the Declaration of Helsinki, and approved by the Ethics Committee of UZ Brussel/VUB (1432022000115, approved on 5 July 2022). This is highlighted in lines 360-362.
We added ‘after their informed consent’ in line 256.
Reviewer 2 Report
No negative comments
Well-designed study and interesting results
Author Response
We thank this reviewer for taking the time and effort to review our manuscript.
Reviewer 3 Report
This is a well-written paper by Vanstokstraeten and colleagues presenting the results of comparison between the vaginal and endometrial microbiota with use of culturomics-based approach.
Author documented presence of particular microbiota species in vagina and endometrium. Over 100 bacterial strains were detected in biological samples. It is worth noted that researchers employed a high-throughput/extensive culture methodology - Culturomics (WASPLab®-assisted culturomics) by combining different agar plates, enrichment broths, and (an)aerobic incubation conditions to cultivate virtually all viable microbiota. This standardized methodology enabled high traceability and reproducibility. Finally, cultivated bacterial samples were analyse with time-of-flight mass spectrometry (MALDI-TOF MS) and full-gene 16S rRNA sequencing.
Authors concluded, that there is a high similarity between the vaginal and endometrial micro-biota based on culturomics performed on paired samples. Nevertheless, authors also found unique species in both the vaginal and endometrial microbiota thus, a unique endometrial microbiome could exist and be of clinical relevance. This is an interesting study, methodologically and clinically valuable. The study was well performed. The authors extensively discussed the results. The idea and results presented in this manuscript may attract attention of researchers studying the role of microbiota in human physiology and pathophysiological conditions (e.g. cancer).
Minor comments:
Comment 1. I am curious, if it is possible to calculate, what is an absolute bacteria number per sample (cm2, ml or other unit) in vagina, cervix and endometrium. Does cervix microbiota represent vaginal or rather endometrial bacteria population pattern, or mixed?
Comment 2. I can see 3 groups of patients within examined 10 patients, 1 group pateints with 5.8-10.53% of common species for vagina and endometrium, 2 group 24-33.33%, 3 group 43-46%. Do these common species differ among groups? Please write a short comment on this issue in the paper.
Comment 3. It would be interesting for readers to know whether unique endometrial bacteria populations differ among 10 patients. Please write a short comment on this issue in results and discussion.
Comment 4. What could be impact of bacterial viruses-bacteriophages on modulation of microbiome composition in vagina and endometrium. Please write a short comment on this issue in the paper.
Comment 5. Is there any association between uterine anatomopathological status and its microbiome representation?
Taken together, this manuscript by Vanstokstraeten and colleagues represents a worthwhile contribution to the human microbiome research. I recommend the manuscript for further publication process.
Author Response
We thank this reviewer for taking the time and effort to review our manuscript.
Minor comments:
Comment 1. I am curious, if it is possible to calculate, what is an absolute bacteria number per sample (cm2, ml or other unit) in vagina, cervix and endometrium. Does cervix microbiota represent vaginal or rather endometrial bacteria population pattern, or mixed?
Unfortunately, we cannot generate quantitative data with culturomics. As a result, it is not possible to use this technique to quantify microorganisms in a sample. This issue is mentioned in lines 145-147 and line 241.
In this study, no cervical samples were analyzed with culturomics. However, based on the available sequence-based literature, the cervical microbiota is indeed very similar (qualitative and quantitative) to that of the vagina. https://doi.org/10.1016/j.fertnstert.2018.04.041
Comment 2. I can see 3 groups of patients within examined 10 patients, 1 group pateints with 5.8-10.53% of common species for vagina and endometrium, 2 group 24-33.33%, 3 group 43-46%. Do these common species differ among groups? Please write a short comment on this issue in the paper.
Thank you for this interesting suggestion. However, we do not observe an association between the species found in the groups you suggested. We also think the sample size in this proof-of-concept is far too small to make any statements about this. In addition, we are only dealing with qualitative data here, which makes such associations even more difficult to make. Therefore, we kindly suggest not elaborating on this issue in this manuscript.
Comment 3. It would be interesting for readers to know whether unique endometrial bacteria populations differ among 10 patients. Please write a short comment on this issue in results and discussion.
Indeed, at the species level, very different bacterial populations were observed among the 10 patients. We agree with this suggestion and added lines 122-126 in the results.
Comment 4. What could be impact of bacterial viruses-bacteriophages on modulation of microbiome composition in vagina and endometrium. Please write a short comment on this issue in the paper.
We agree with this suggestion. We added a short comment on this issue in lines 210-213 in the discussion and added an extra citation.
Comment 5. Is there any association between uterine anatomopathological status and its microbiome representation?
Despite we mention the anatomopathological status of the patients, the sample size in this study is too small to make a statement on this. Although sequence-based papers already discussed this issue, solid correlations between endometrial anatomopathological status and bacterial composition have not been established, yet.
Taken together, this manuscript by Vanstokstraeten and colleagues represents a worthwhile contribution to the human microbiome research. I recommend the manuscript for further publication process.